# Isolation and identification of hyaluronan-degrading bacteria from Japanese fecal microbiota

Hazuki Akazawa[1], Itsuko Fukuda[1,2]*, Haruna Kaneda[3], Shoichi Yoda[3], Mamoru Kimura[3], Ryohei Nomoto[4], Shuji Ueda[1], Yasuhito Shirai[1], Ro Osawa[2,5]

1 Department of Agrobioscience, Graduate School of Agricultural Science, Kobe University, Kobe, Hyogo, Japan, 2 Research Center for Food Safety and Security, Graduate School of Agricultural Science, Kobe University, Kobe, Hyogo, Japan, 3 R&D Division, Kewpie Corporation, Sengawa Kewport, Chofu-shi, Tokyo, Japan, 4 Department of Infectious Diseases, Kobe Institute of Health, Chuo-ku, Kobe, Hyogo, Japan, 5 Department of Bioresource Science, Graduate School of Agricultural Science, Kobe University, Nada-ku, Kobe, Hyogo, Japan

* itsuko@silver.kobe-u.ac.jp

**Data Availability Statement:** The completed and draft genome sequences the two strains described in this study have been deposited in DDBJ/EMBL/GenBank. The URLs of the completed and draft

## Abstract

Hyaluronan (HA) is a high-molecular-weight glycosaminoglycan and widely distributed in all connective tissues and organs with diverse biological functions. HA has been increasingly used as dietary supplements targeted to joint and skin health for humans. We here first report isolation of bacteria from human feces that are capable of degrading HA to lower molecular weight HA oligosaccharides (oligo-HAs). The bacteria were successfully isolated via a selective enrichment method, in which the serially diluted feces of healthy Japanese donors were individually incubated in an enrichment medium containing HA, followed by the isolation of candidate strains from streaked HA-containing agar plates and selection of HA-degrading strains by measuring HA using an ELISA. Subsequent genomic and biochemical assays identified the strains as *Bacteroides finegoldii*, *B. caccae*, *B. thetaiotaomicron*, and *Fusobacterium mortiferum*. Furthermore, our HPLC analysis revealed that the strains degraded HA to oligo-HAs of various lengths. Subsequent quantitative PCR assay targeting the HA degrading bacteria showed that their distribution in the Japanese donors varied. The evidence suggests that dietary HA is degraded by the human gut microbiota with individual variation to oligo-HAs components, which are more absorbable than HA, thereby exerting its beneficial effects.

## Introduction

Hyaluronan (HA) is a major glycosaminoglycan of the extracellular matrix, and is a linear polymer of a repeating disaccharide, β1–3D-*N*-acetylglucosamine (GlcNAc) and β1–4D-glucuronic acid (GlcUA) with molecular weight from tens of thousands to millions. HA exists ubiquitously in the extracellular matrix in vertebrates and is particularly abundant in connective tissues and organs in the body, including skin, synovial fluid, blood vessels, brain, cartilage

genome sequences are as below: Bacteroides finegoldii JCM 13345 (chromosome, plasmid) https://www.ncbi.nlm.nih.gov/nuccore/AP027244 https://www.ncbi.nlm.nih.gov/nuccore/AP027245 Bacteroides finegoldii strain KUBF1 (draft sequence) https://www.ncbi.nlm.nih.gov/Traces/wgs/BSFS01?display=contigs.

**Funding:** This study was supported by Japan Innovative Bioproduction Kobe. The funders had no role in study design, data collection and analysis, decision to publish, or preparation of the manuscript.

**Competing interests:** The authors have declared that no competing interests exist.

**Abbreviations:** KUHIMM, Kobe University Human Intestinal Microbiota Model; HA, hyaluronan; HPLC, high-performance liquid chromatography; PCR, polymerase-chain reaction.

[1]. The safety of HA as a food has been confirmed, and it has been used to various dietary supplements [1,2]. For example, a skin-improving effect was confirmed by ingesting 240 mg/day HA for 6 weeks [2], but the detailed mechanism of how HA with such high molecular weight exert it beneficial effects is yet to be elucidated.

An in vivo study [3] showed that approximately 90% of orally administered HA was detected by autoradiography in plasma of rats using $^{14}$C-HA and suggested that the HA was somehow absorbed, metabolized, and transferred to the skin. Meanwhile, an in vitro study [4] demonstrated that HA was not degraded by artificial gastric and intestinal juices, but by the rat cecal microbiota to HA oligosaccharides (oligo-HAs), which was absorbed through a pseudo-intestinal wall. As for bacterial HA degradation, it has been reported elsewhere [5,6] that bacterial strains belonging to the genus *Bacteroides*, *Prevotella*, *Porphyromonas*, and *Fusobacterium*, which are common inhabitants of human gut, are capable of degrading HA. Little is, however, known about degradation patterns of HA by the gut bacteria. We here report isolation and identification of bacteria that degrade HA to oligo-HAs and their distribution in the microbiota of 10 healthy human donors.

## Materials and methods

### Materials

Gifu anaerobic medium (GAM) was from Nissui Pharmaceutical Co. (Tokyo, Japan). HA (MW around 300,000) was from Kewpie Corporation. All other reagents used were the highest grade of the commercially available.

### Preparation of human fecal innoculum

Fecal samples from 10 donors were transported anaerobically right after collected, and diluted 10-fold with PreserWell (MPR, Miyagi, Japan) within 3 hours, then stored at −80°C until use. The study was performed in accordance with the guidelines of Kobe University Hospital and approved by the institutional ethics review board of Kobe University (Approval Number: 1902), and the informed written consent was obtained from each healthy donors or parents in case of minors.

### Kobe University Human Intestinal Microbiota Model in Bottle (KUHIMMiB)

We used the KUHIMMiB, which was miniaturized from KUHIMM [3,4]. Twenty mL of GAM semisolid without dextrose (Nissui Pharmaceutical) from which agar was removed using filter paper (Grade 2, GL Sciences, Tokyo, Japan) was dispensed into each 50 mL volume vial (NICHIDEN-RIKA GLASS, Hyogo, Japan). The vial was covered with a butyl rubber stopper (NICHIDEN-RIKA GLASS) and an open top crimp cap (GL Sciences), and strictly sealed with a hand crimper (OSAKA CHEMICAL, Osaka, Japan). After degassing for 10 min using a vacuum pump (ULVAC, Kanagawa, Japan), aerate the mixed gas of $N_2$ (80%): $CO_2$ (18%): $H_2$ (2%) at 0.1 MPa for 15 sec then repeated 3 times to make the inside of the vial anaerobic. The medium was sterilized in an autoclave at 115°C for 15 min. To start the fermentation, 50 μL of the supernatant of thawed cryopreserved fecal inoculum centrifuging at 60 g for 1 min was inoculated through a butyl rubber stopper by an injection syringe and a syringe needle and cultured at 37°C. For isolation and identification of HA-degrading bacteria, 200 μL of 10 mg/mL HA was added to the medium. Aliquots of the fermentation cultures were collected at 30 h after the inoculation, and stored at −20°C until subjection to detection of HA.

## Detection of HA

HA remained in the medium was determined by an ELISA kit (R&D Systems, Inc., MN) according to the manufactural protocol. Briefly, the supernatant without the bacterial inoculum was diluted 4,000-fold, and the culture supernatant with the bacterial inoculum was diluted 200-fold to be within the detection range of the ELISA kit, and after colorization, absorbance at 450 nm was measured in a plate reader.

## Isolation of HA-degrading bacteria

To isolate the bacteria degrading HA from fecal microbiota, 150 μL of GAM without dextrose medium containing 100 μg/mL HA was dispensed to all well of a 96-hole V-bottom microplate, and additional 40 μL of 100 μg/ml HA-added GAM and 10 μL of fecal inoculum from 4 donors who efficiently degrading HA was added to the first row of the plate to make a total volume of 200 μL. Next, an aliquot of 50 μL was aspirated from this first row and dispensed into the wells of the second row of the plate, mixed well and diluted, and the same procedure was performed for subsequent rows. By this operation, the sample was diluted 4-fold to $4^{11}$-fold. The plates were then incubated anaerobically using AnaeroPack (Mitsubishi Gas Chemical Co. Inc., Tokyo, Japan) at 37˚C for 24 h and centrifuged to obtain cell-free spent media from each well. HA remained in the medium was determined by ELISA as described above. The bacterial suspension of the well with the highest intestinal content dilution exhibiting apparent HA degradation by ELISA was selected for each of the 5 samples to isolate HA–degrading bacteria as follows: the bacterial suspensions of the 5 samples were streaked on HA-added GAM agar plates using platinum loops. The agar plates were then anaerobically incubated (Mitsubishi Gas Chemical) at 37˚C for 24 h. After incubation, 5–10 well-separated colonies on each plate were randomly selected and screened for HA degradation using ELISA.

## DNA extraction

DNA extraction was performed according to Marmur's procedure [5]. Briefly, 200 μL from the culture medium was added into a screwed freestanding microtube containing 300 mg glass beads (0.1 mm diameter). 500 μL of TE (10 mM Tris-HCl, 1 mM EDTA, pH 8.0) saturated phenol, 250 μL of TE buffer, and 50 μL of 10% sodium dodecyl sulfate was also dispensed into the above microtubes and mixed. The mixed solution was then shaken for 30 seconds using Fast Prep-24 (MP Biomedicals SARL, Illkirch, France). The supernatant of centrifugation at 17,120 g for 5 min was extracted with 400 μL of a phenol-chloroform-isoamyl alcohol (25:24:1) mixture, and then centrifuged at 17,120 g for 5 min. The supernatant 250 μL was collected in microtubes aliquoted with 275 μL of isopropanol and 25 μL of 3 M sodium acetate and allowed to stand at -20˚C for 10 to 15 minutes. The extracted DNA precipitates were collected by centrifugation at 17,120 g for 5 min, washed with 70% ethanol, and air-dried for 10–30 min. The extracted DNA was then dissolved in TE. The above DNA extracts were stored at -20˚C until use.

## Identification of HA degrading bacteria

To identify of HA degrading bacteria, PCR was performed using the Bacterial rDNA PCR kit (TaKaRa Bio Inc., Shiga, Japan) according to the protocol, using the genomic DNA obtained by the above operation. The base sequence of the obtained PCR product was determined using four types of primers: Sequencing Primer F1 (bacterial), Sequencing Primer F2 (bacterial), Sequencing Primer R1 (bacterial), and Sequencing Primer R2 (bacterial). A sequence reaction was performed according to ABI Big Dye v3.1, and the base sequence was determined by a

**Table 1. List of primer sequences and the amplicon sizes for qPCR.**

| Target | Sequence (5′ to 3′) | Amplicon size (nt) |
|---|---|---|
| All eubacteria | ACTCCTACGGGAGGCAGCAGT | 200 |
| | GTATTACCGCGGCTGCTGGCAC | |
| *B. finegoldii* | ACCTGATGGCATAGGATTATCGC | 223 |
| | GCTGGTTCAGACTCCCGTCC | |
| *B. caccae* | TACCTCATACTCGGGGATAGC | 149 |
| | GTAGTCTTGGTGGGCCGTTA | |
| *B. thetaiotaomicron* | CCCGATAGTATAATGAAACCGC | 207 |
| | CGCCCATTGACCAATATTCC | |
| *F. mortiferum* | GCTGCTGGCACGTATTTAGC | 250 |
| | GTGAGGTAACGGCTCACCAA | |

16-line capillary DNA sequencer (ABI3130xl Genetic Analyzer). The obtained nucleotide sequence was searched for homology by BLAST of NCBI.

## Quantification of HA degrading bacteria by quantitative PCR (qPCR)

qPCR was performed targeting all eubacteria and HA degrading bacteria, including *Bacteroides finegoldii*, *B. caccae*, *B. thetaiotaomicron*, and *Fusobacterium mortiferum* as shown in Table 1.

## Analysis of oligo-HAs by high-performance liquid chromatography (HPLC)

The degraded products of HA, oligo-HAs were analyzed by HPLC as described previously [6]. Because GAM was not appropriate to detection of oligo-HAs due to admixture, minimum nutritional medium for *Bacteroides* [7] was used for incubation of HA with HA-degrading bacteria, and the supernatant of incubated medium was applied to HPLC as previously described [6].

## Analysis of complete genome sequence of *B. finegoldii*

The type strain of *B. finegoldii* JCM13345[T] and the isolated strain were cultured in GAM under the anaerobic condition, and the genome DNA was extracted by Wizard HMW DNA Extraction Kit (Pomega Corp., WI) according to the manufactural protocols, and the complete genome sequence was analyzed by the long-read sequencer using PacBio RSII at Macrogen Japan Corp. (Tokyo, Japan). The Average Nucleotide Identity (ANI) value between type strain and isolated strain have been calculated by using ANI calculator (http://enve-omics.ce.gatech.edu/ani/) [8]. The completed and draft genome sequences the two strains have been deposited in DDBJ/EMBL/GenBank under BioProject accession no. PRJDB15142. The accession numbers of complete genome sequences of type strain are AP027244-AP027245, and draft genome sequence of isolated strain are BSFS01000001- BSFS01000023.

## Results

### Fecal microbiota possess HA-degrading ability

To isolate and identify an HA-degrading bacteria, fecal microbiota from healthy 10 donors were subjected to KUHIMMiB containing 100 μg/mL HA, resulting in the fecal microbiota tested here degraded HA to under the detection limit of ELISA (0.625 ng/mL) in 9 donors

**Table 2. Ability of HA degradation in fecal microbiota from 10 donors.**

| Fecal donor | Remained HA (%) | | |
|---|---|---|---|
| 1 | 1.6 | ± | 2.8 |
| 2 | 0.0 | ± | 0.0 |
| 3 | 8.0 | ± | 1.7 |
| 4 | 4.9 | ± | 4.5 |
| 5 | 0.9 | ± | 1.6 |
| 6 | 8.2 | ± | 1.9 |
| 7 | 1.6 | ± | 2.6 |
| 8 | 3.3 | ± | 4.3 |
| 9 | 2.2 | ± | 3.3 |
| 10 | 1.1 | ± | 0.9 |

Remained HA (%) was measured after fermentation for 9 h with each fecal donor. Data were shown as the mean ± standard deviation (n = 3) of the values obtained after the independent experiments.

except for donor 8 after fermentation for 30 h. Therefore, the ability of HA degradation was examined after fermentation for 9 h in 10 fecal microbiota (Table 2). It was found that the fecal microbiota from donor 2 degraded HA to 0.00±0.00% most efficiently among 10 donors.

## *B. finegoldii* as an HA-degrading bacteria from human fecal microbiota

The fecal microbiota from donor 2 were then subjected to stepwise dilution from 4-fold to $4^{11}$-fold, and fermented with 100 μg/mL HA. The isolated single colony was identified as *Bacteroides finegoldii*, which has not been reported as HA-degrading bacteria. The isolated strain of *B. finegoldii* was found to be 99.2% homologous to its 16S rRNA gene of type strain by NCBI BLAST search. Comparison of the isolated strain with the type strain JCM13345[T] showed no difference in colony formation on plain medium and gram staining results between the standard and wild strains (Fig 1). Rapid ID32A Api also revealed that there was no difference between these two strains (Table 3). Whole genome sequencing of the isolated and type strains of *B. finegoldii* resulted in 23 contigs in the isolated strain and 2 contigs in the type strain as a result of de novo genome assembly of the PacBio data. Since there were many contigs in the isolated strain, we decided to proceed with the whole genome sequence of the type strain. The ANI value calculated by comparing the genome sequence data of the isolate and the standard

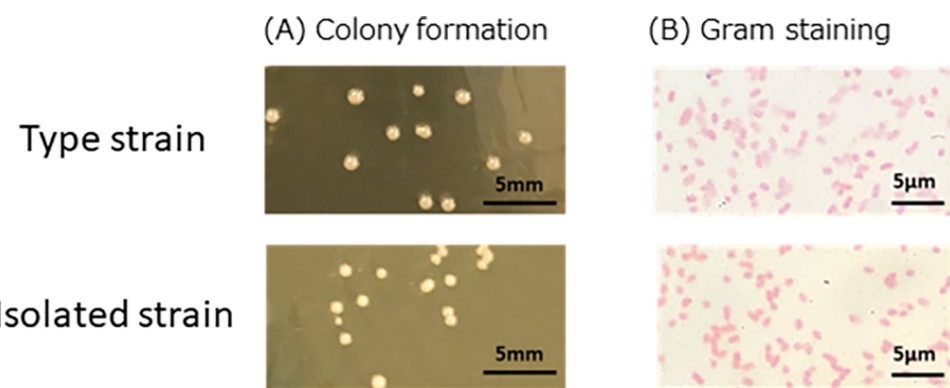

**Fig 1. Basic characteristics of type strain and isolated strain of *B. finegoldii*.** (A) Colony formation on GAM agar medium and (B) Gram staining were observed after 24 h-incubation in anaerobic condition.

**Table 3. Comparison of characteristics of the type strain and isolated strain of *Bacteroides finegoldii* from human fecal microbiota.**

| Strain | URE | ADH | αGAL | βGAL | βGP | αGLU | βGLU | αARA | βGUR | βNAG | MNE | RAF | NIT | IND | PAL | ArgA | ProA | LGA | PheA | LeuA | PyrA | TyrA | AlaA | GlyA | GDC | αFUC | HisA | GGA | SerA |
|---|---|---|---|---|---|---|---|---|---|---|---|---|---|---|---|---|---|---|---|---|---|---|---|---|---|---|---|---|---|
| JCM13345^T | - | - | + | + | - | + | + | + | - | + | + | + | - | - | + | - | - | + | - | - | - | - | + | - | + | - | - | - | - |
| Isolated | - | - | + | + | - | + | + | + | - | + | + | + | - | - | + | - | - | + | - | - | - | - | + | - | + | - | - | - | - |

The characteristics of both strains were determined by Rapid ID32A Api. Results were shown as +, positive; −, negative.

Abbreviations of reactions/enzymes are as follows: URE, urease; ADH, arginine dihydrolase; αGAL, α-galactosidase; βGAL, β-galactosidase; βGP, β-galactosidase-6-phosphate; αGLU, α-glucosidase; βGLU, β-glucosidase; αARA, α-arabinosidase; βGUR, β-glucuronidase; βNAG, N-acetyl-β-glucosaminidase; MNE, mannose; RAF, raffinose; NIT, nitrate reduction; IND, indole production; PAL, phosphatase alkaline; ArgA, arginine arylamidase; ProA, proline arylamiase; LGA, leucylglycine arylamidase; PheA, phenylalanine arylamidase; LeuA, leucine arylamidase; PyrA, pyroglutamic acid arylamidase; TyrA, tyrosine arylamidase; AlaA, alanine arylamidase; GlyA, glycine arylamidase; GDC, glutamic acid decarboxylase; αFUC, α-fucosidase; HisA, histidine arylamidase; GGA, glutamylglutamic acid arylamidase; SerA, serine arylamidase.

strain was 98.33%, which is above the threshold (95%) for being considered the same species, and thus we concluded that the isolate was *B. finegoldii*. The contigs of the type strain were sequenced in order of length (4,921,693 bp and 30,872 bp) as sequence1 and sequence2, respectively, and were confirmed to have a circular structure.

## Determination of *B. finegoldii* in fecal microbiota from 10 donors

To confirm the HA-degrading ability of fecal microbiota owes to the cell number of *B. finegoldii*, the bacterial cell number of all eubacteria and *B. finegoldii* in fecal microbiota from 10 donors were determined. As a result, donor 2 contained $10^{10.01}$ cells/mL of *B. finegoldii* with the strongest HA-degrading ability, expectedly, whereas donors 3, 4, 6, and 8 showed poor ability to degrade HA with relatively small amount of *B. finegoldii* (Tables 2 and 4). Donors 1, 5, and 7 showed degrading-HA ability with the remained HA of 0.9–1.6% (Table 2), although these microbiota contained less than $10^{7.30}$ cells/mL of *B. finegoldii*. These results suggested that microbiota from donors 1, 5, and 7 contained the other HA-degrading bacteria than *B. finegoldii*. Therefore, we proceeded to isolate HA-degrading bacteria other than *B. finegoldii*.

## The other HA-degrading bacteria than *B. finegoldii*

Similar to the isolation of *B. finegoldii*, we isolated HA-degrading bacteria from the fecal microbiota that had highly HA-degrading ability, *F. mortiferum* from donor 1, *B. caccae* from donor 5, and *B. thetaiotaomicron* from donor 7. In this study, *B. caccae*, *B. thetaiotaomicron*, and *F. mortiferum* were isolated and identified from the fecal microbiota in addition to *B. finegoldii* (Fig 2) as HA-degrading bacteria. The distribution of these strains in the fecal microbiota

**Table 4. Distribution of HA-degrading bacteria in fecal microbiota of 10 healthy Japanese donors.**

| Fecal donor | Bacterial cell number | | | | |
|---|---|---|---|---|---|
| | All | *B. finegoldii* | *B. caccae* | *B. thetaiotaomicron* | *F. mortiferum* |
| 1 | 8.87±0.01 | 6.19±0.01 | 7.08±0.01 | UDL | 8.54±0.01 |
| | 11.14±0.00 | 7.24±0.77 | 9.33±0.00 | 6.79±0.22 | 10.99±0.00 |
| 2 | 8.99±0.01 | 8.25±0.02 | 7.87±0.01 | UDL | 8.90±0.02 |
| | 11.06±0.01 | 10.01±0.00 | 9.57±0.02 | 6.72±0.37 | 10.93±0.00 |
| 3 | 8.62±0.01 | 6.12±0.02 | 6.21±0.00 | UDL | 8.37±0.00 |
| | 11.01±0.01 | 8.02±1.04 | 8.69±0.02 | 5.89±0.51 | 10.20±0.01 |
| 4 | 8.57±0.01 | 6.37±0.05 | 7.30±0.00 | 6.36±0.01 | 7.96±0.00 |
| | 10.94±0.00 | 7.55±0.72 | 9.40±0.00 | 8.75±0.00 | 9.84±0.01 |
| 5 | 8.42±0.05 | 6.12±0.09 | 6.88±0.00 | 5.82±0.02 | 7.84±0.01 |
| | 10.94±0.00 | 7.30±0.57 | 9.26±0.00 | 8.09±0.00 | 10.69±0.00 |
| 6 | 8.24±0.03 | 6.11±0.09 | 5.20±0.00 | 5.08±0.21 | 7.81±0.00 |
| | 10.98±0.00 | 6.46±0.03 | 6.88±0.24 | 5.76±0.35 | 9.63±0.01 |
| 7 | 8.35±0.01 | 6.31±0.04 | 6.59±0.00 | UDL | 8.11±0.00 |
| | 10.99±0.00 | 6.59±0.15 | 9.08±0.03 | 7.36±0.25 | 10.14±0.01 |
| 8 | 8.46±0.03 | 6.11±0.09 | 6.45±0.01 | 5.89±0.03 | 8.13±0.00 |
| | 11.14±0.01 | 6.58±0.09 | 8.74±0.03 | 8.16±0.00 | 9.82±0.02 |
| 9 | 8.88±0.01 | 7.56±0.02 | 7.92±0.00 | 5.50±0.18 | 8.86±0.01 |
| | 11.00±0.01 | 9.24±0.00 | 9.68±0.01 | 7.26±0.19 | 10.24±0.00 |
| 10 | 8.70±0.01 | 6.37±0.04 | 7.53±0.01 | 5.13±0.06 | 8.48±0.01 |
| | 11.11±0.00 | 6.99±0.75 | 9.17±0.01 | 6.93±0.24 | 10.96±0.01 |

Upper rows and the lower rows in each columns show the bacterial cell numbers before fermentation and that of after fermentation after 9 h, respectively. Bacterial cell numbers were shown as $Log_{10}$ cells/mL. UDL, under the detection limit ($<5$). Data were shown as the mean ± standard deviation (n = 3) of the values obtained after the independent experiments.

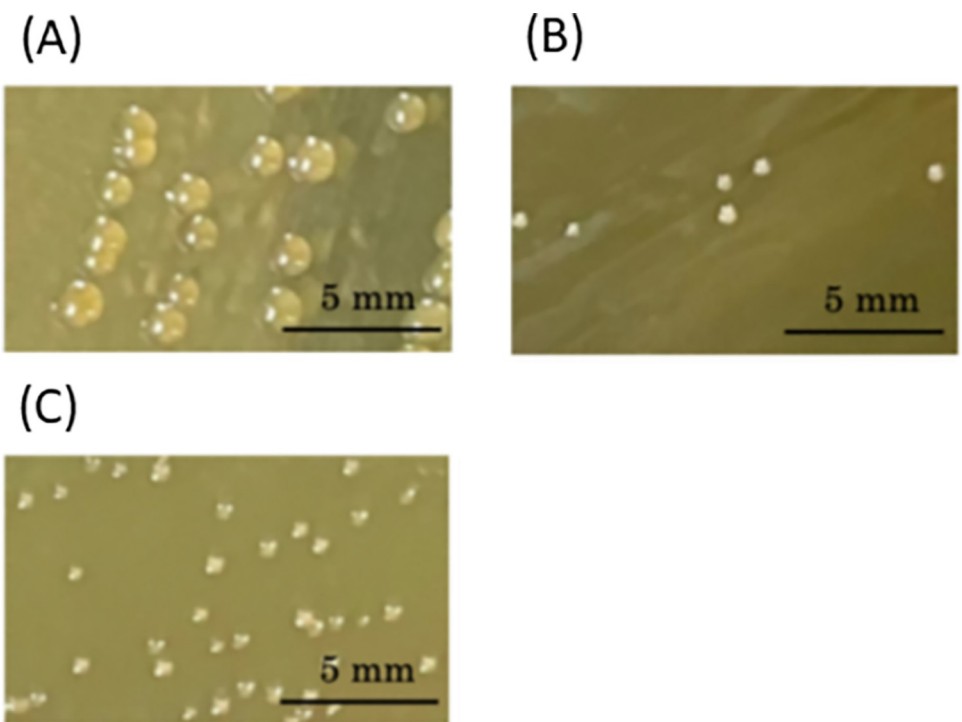

**Fig 2. Colony formation.** Colony formation of GAM agar medium of (A) *F. mortiferum*, (B) *B. caccae*, and (C) *B. thetaiotaomicron* was observed after 24 h-incubation in anaerobic condition.

of 10 donors were again confirmed (Table 4). *B. finegoldii* and *B. caccae* were abundant in donors 2 and 9 with high degradation ability, and *B. caccae* and *B. thetaiotaomicron* were present in the other microbiota, indicating that these bacteria also affect the HA-degrading ability. Regarding with donors 3 and 6, where HA-degrading bacteria of the *Bacteroides* spp. were scarce, HA residuals were 8.0% and 8.2%, respectively (Tables 2 and 4), suggesting that the contribution of the genus *Bacteroides* to HA degradation was significant regardless of the presence of *F. mortiferum*. In addition, despite the low abundance of *Bacteroides* spp. and *F. mortiferum* in donors 4 and 8, the remained HA were 4.9% and 3.3%, respectively, suggesting the presence of other HA-degrading bacteria in addition to the four species found in this study.

## Production of oligo-HAs by a single bacterium

In the human fecal microbiota, the HA-degrading bacteria found in this study were present in various proportions, suggesting that their presence ratio may affect HA degradation in the intestine. To confirm the degradation of HA in a single bacterium, *B. finegoldii* was cultured on minimal HA-containing medium and analyzed by HPLC. As shown in Fig 3, *B. finegoldii* degraded HA in a time-dependent manner, and oligo-HAs including disaccharides, tetrasaccharides, hexasaccharides, and octasaccharides were produced, whereas HA was slightly degraded. Peak area of oligosaccharides was ordered as tetra-, hexa-, octa-, and disaccharides. The other HA-degrading bacteria were also cultured on minimal HA-containing medium and oligo-HAs were detected. Although the cell numbers of these bacteria could not be controlled, it was confirmed that oligo-HAs were produced also by *F. mortiferum*, *B. thetaiotaomicron*, *B. caccae* (Fig 4).

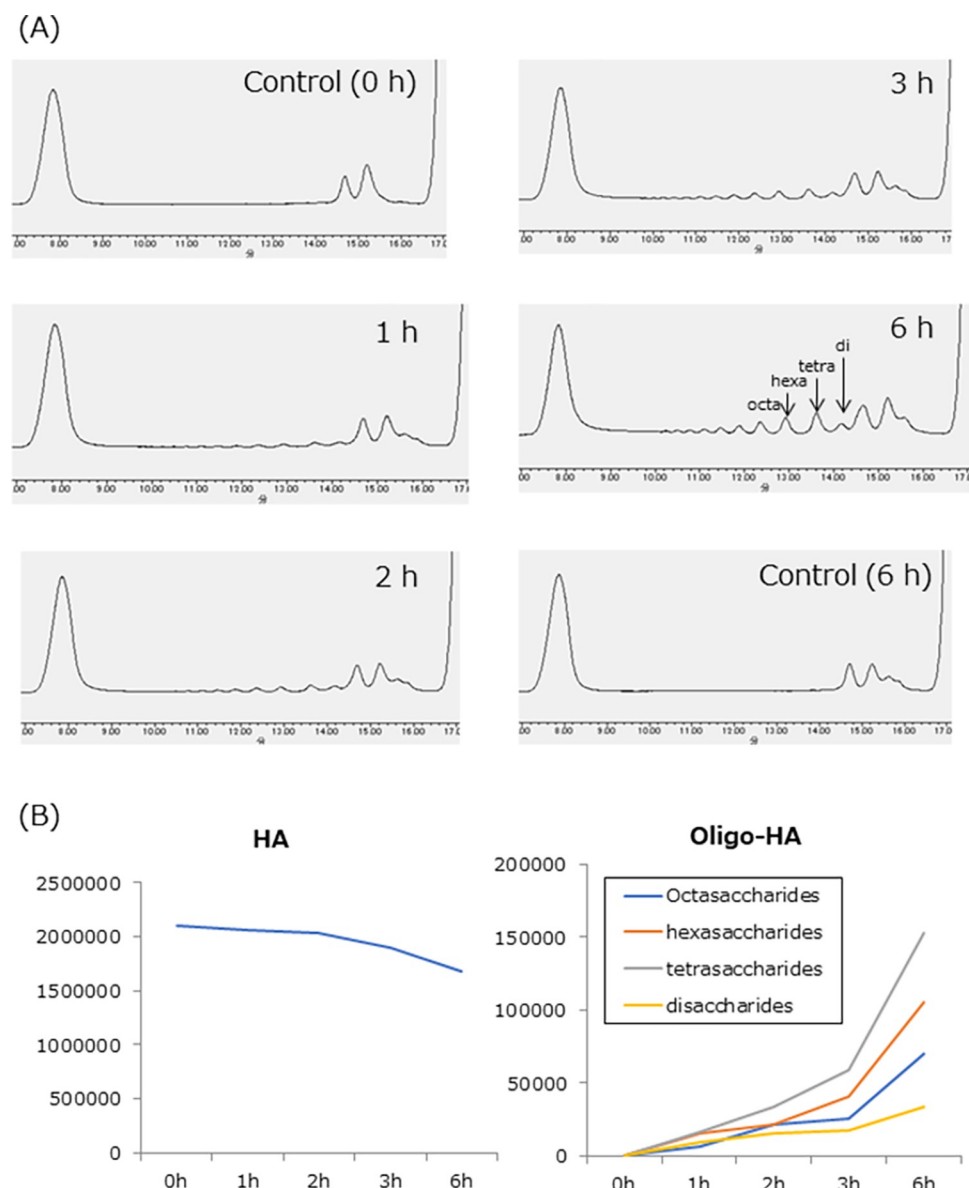

**Fig 3. Detection of oligo-HAs after fermentation of HA with *B. finegoldii*.** (A) Representative HPLC chromatograms of oligo-HAs and HA 1, 2, 3, 6 h after fermentation with or without *B. finegoldii* were shown. (B) Peak area of HA and oligo-HAs were plotted.

## Discussion

In this study, we isolated and identified *B. finegoldii*, *B. caccae*, *B. thetaiotaomicron*, and *F. mortiferum* as HA-degrading bacteria from human fecal microbiota from Japanese donors. Because we isolated these bacteria using KUHIMMiB, which is the appropriate model for intestinal microbiota [3,4], it was suggested that these bacteria are dominant species and degrade HA in human large intestine. HA at 100 µg/mL was degraded to below the detection limit in almost donors tested here in KUHIMMiB after 30 h, and this concentration is corresponding to a daily intake of 15 mg [9]. Because the molecules approximately up to 4,000 Da with diameter of 20 Å is considered to permeate through the intestinal cells [10,11], high

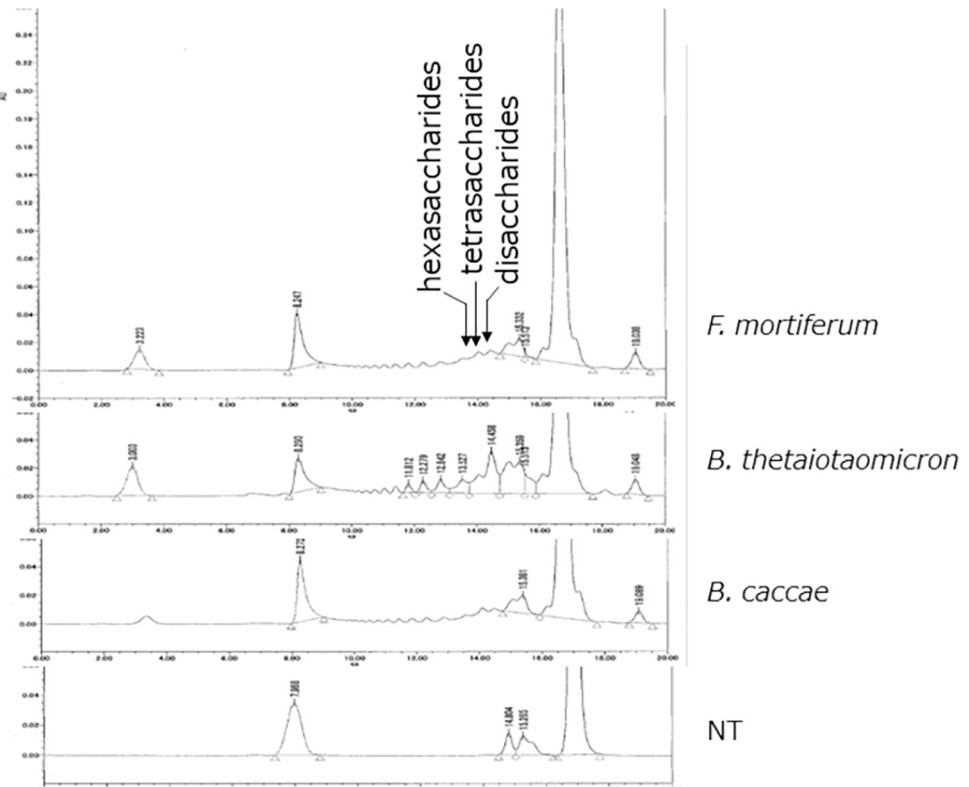

**Fig 4. Detection of oligo-HAs after fermentation of HA with *F. mortiferum*, *B. thetaiotaomicron*, and *B. caccae*.**
Representative HPLC chromatograms of oligo-HAs and HA 24 h after fermentation with *F. mortiferum*, *B. thetaiotaomicron*, and *B. caccae* were shown. NT, no treatment.

molecular HA (MW around 300,000) probably reaches to the large intestine. This suggests that HA is degraded by the intestinal microbiota after reaching the large intestine. Although Balogh et al. [12] reported orally administered high molecule HA rapidly incorporated to the body through lymphatic pathway, 86.7–95.6% of radioactivity was recovered in urine and feces in the same report, indicating that the most of the orally administered high molecule HA go through the intestinal tract.

Since it has been reported that HA is degraded by the contents of rat cecum [6], the degradation of HA was seemed to be caused by intestinal bacteria, and HA-degrading bacteria were isolated from human fecal microbiota with high activity of HA degradation in this study. Regardless of the cell number of *F. mortiferum*, a lower total number of the three *Bacteroides* species compared to the others resulted in about 8% HA remaining in the donor even after 9 h-fermentation. The intestinal tract contains trillions of bacteria, each of which shares metabolites [13]. Therefore, although *F. mortiferum* is active in HA degradation when cultured as a single bacterium in HA-containing medium, its activity is low in the human intestine, where a variety of substances are present, suggesting that these *Bacteroides* bacteria are actually involved in HA degradation. This suggests that the rate of HA degradation in humans varies among individuals, and that the higher the number of bacteria of these three *Bacteroides* species, the faster the rate of HA degradation.

In this study, we found that HA degraded into oligosaccharides level by human intestinal HA-degrading bacteria. Although several reports indicate that intestinal microbiota have

hyaluronidases and utilize HA [14–16], this is the first report indicating oligo-HAs are produced by the intestinal HA-degrading bacteria. The MW of oligo-HAs is ranged around 400–1600, which is probably able to permeate through the epithelial cells in the large intestine [10,11]. In vitro studies indicate that oligo-HAs reveal various biological effects; oligo-HAs suppress cytokine expression in a toll-like receptor 4 (TLR4)-dependent manner in macrophages [17], oligo-HAs show anti-inflammatory effects and improves epidermal barrier in keratinocytes [18], and oligo-HAs stimulate matrix metalloproteinase and anabolic gene expression in ovine intervertebral disc cells [19]. Ghatak et al. demonstrated that oligo-HAs inhibit growth of tumor cells through suppressing the phosphoinositide 3-kinase/Akt pathway in vitro and in vivo [20]. These previous reports suggest that oligo-HAs are the appropriate molecular size to ensure the functionality of HA in the target cells. From this perspective, HA-degrading bacteria that produce oligo-HAs can be considered as probiotics. We have demonstrated that HA-degrading bacteria count vary among individuals, therefore, the biological activity of orally administered HA might differ depending on the strains and its proportion of HA-degrading bacteria in the individuals. The bioavailability of oligo-HA produced by HA-degrading bacteria, including its capitalization by the other bacterial groups in the large intestine, should be determined in both KUHIMM and human intervention studies, in the future.

*B. finegoldii*, which was isolated from the fecal microbiota with the highest degradation ability, has not been reported as HA-degrading bacteria [21]. Since the whole genome sequence of *B. finegoldii* was not yet available, we attempted to analyze the whole genome sequence of the type strain and the isolated strain. To verify whether the annotated ORFs contain HA-degrading enzymes, we searched for candidate genes in the type strain using the sequence of Query known HA-degrading enzymes as a reference in protein blast, and found that β-hexosaminidase was identified as a candidate HA-degrading enzyme. Pfam domain search of this candidate gene, BFINE_10010, revealed that this enzyme was predicted to be β-N-acetylhexosaminidase belonging to enzyme number EC:3.2.1.52. Although their homology is low, about 30% in parts, the domain that degrades N-acetylglucosamine was confirmed to be conserved. Since it is predicted to be difficult to create a strain of *B. finegoldii* deficient in HA-degrading enzymes, we would like to further validate the candidate genes by, for example, evaluating whether gene expression of the degrading enzymes can be induced by HA-containing media.

Enzymes that hydrolyze HA has been reported as 3 groups: HA glycanohydrolase (EC:3.2.1.35), HA glycanohydrolase (EC:3.2.1.36), and HA lyase (EC:4.2.2.1, formerly EC:4.2.99.1) [14]. In this study, we found beta-N-acetylhexosaminidase (EC:3.2.1.52) as a candidate of HA-degrading enzyme of *B. finegoldii*. The reaction of glycosaminoglycan degradation of this enzyme is a unique pathway, which has not been reported as the HA hydrolyzation mechanism. The HA-degrading activity of *B. finegoldii* is relatively higher than that of the other bacteria, indicating that the enzyme seemed to be secreted outside the cells as well as the other HA-degrading bacteria [15]. The further studies would be needed to clarify the characteristic of the unique enzyme of *B. finegoldii*.

High molecular weight (900 kDa) of HA has been reported to control immune system via toll-like receptor 4 in the human intestinal epithelial cell line without degradation [22], whereas up to 35 kDa HA has been reported to decrease bacterial infection, increase the expression of a tight junction protein, and reduce intestinal permeability [23]. These reports suggest that molecular size of HA affect the biological activity, including absorption rate and/or target receptors. If this is the case, this study points to the possibility that the functionality of dietary HA depends on the population size of the HA degrading bacteria in the gut microbiota of the host, varying from person to person.

## Author Contributions

**Conceptualization:** Mamoru Kimura, Ro Osawa.

**Data curation:** Itsuko Fukuda.

**Funding acquisition:** Yasuhito Shirai.

**Investigation:** Hazuki Akazawa, Itsuko Fukuda, Haruna Kaneda, Shoichi Yoda, Mamoru Kimura, Ryohei Nomoto, Shuji Ueda.

**Methodology:** Ro Osawa.

**Project administration:** Ro Osawa.

**Resources:** Haruna Kaneda, Shoichi Yoda, Mamoru Kimura.

**Supervision:** Yasuhito Shirai, Ro Osawa.

**Writing – original draft:** Itsuko Fukuda.

**Writing – review & editing:** Yasuhito Shirai, Ro Osawa.

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
