## [Decision Letter · Decision Letter 0]

23 Feb 2023

PONE-D-23-02543Isolation and identification of hyaluronan-degrading bacteria from Japanese fecal microbiotaPLOS ONE

Dear Dr. Fukuda,

Thank you for submitting your manuscript to PLOS ONE. After careful consideration, we feel that it has merit but does not fully meet PLOS ONE’s publication criteria as it currently stands. Therefore, we invite you to submit a revised version of the manuscript that addresses the points raised during the review process.

We look forward to receiving your revised manuscript.

Kind regards,

Awatif Abid Al-Judaibi, PhD

Academic Editor

PLOS ONE

Journal Requirements:

"This study was supported by Japan Innovative Bioproduction Kobe. "

"This study was supported by Japan Innovative Bioproduction Kobe. The funders had no role in study design, data collection and analysis, decision to publish, or preparation of the manuscript."

Reviewers' comments:

Reviewer's Responses to Questions

**Comments to the Author**

1. Is the manuscript technically sound, and do the data support the conclusions?

Reviewer #1: Yes

Reviewer #2: Partly

2. Has the statistical analysis been performed appropriately and rigorously? 

Reviewer #1: N/A

Reviewer #2: N/A

3. Have the authors made all data underlying the findings in their manuscript fully available?

Reviewer #1: Yes

Reviewer #2: Yes

4. Is the manuscript presented in an intelligible fashion and written in standard English?

Reviewer #1: Yes

Reviewer #2: Yes

5. Review Comments to the Author

Reviewer #1: Although, the manuscript is technically sound, but the author failed to justify the sample size. The manuscript was presented in intelligible fashion. However, how the medium was made selective should be clearly explained.

Reviewer #2: The paper paper is an interesting one. However, it needs English editing to make certain statements clearer in order to improve upon the write up. Some of these have been suggested in the file uploaded for consideration.

6. PLOS authors have the option to publish the peer review history of their article (what does this mean?). If published, this will include your full peer review and any attached files.

Reviewer #1: **Yes: **Pelumi Daniel Adewole

Reviewer #2: No

---

## [Author Response · Author response to Decision Letter 0]

28 Mar 2023

Thank you for your constructive comments to our manuscript to PLoS ONE. We have revised the manuscript in accordance with your comments as follows:

1. In lines 15, 23, 49, 252, 442 we have changed “feces of healthy Japanese subjects” to “feces of healthy Japanese donors” as you pointed out.

2. In lines 17-18 and 89-90, we have revised the statement of ELISA.

3. In lines 90-91, the word “absorption” was changed to “absorbance”.

4. We have revised the statement about the step of dilution of fecal inoculum in lines 98-101.

---

## [Editor Report · Decision Letter 1]

3 Apr 2023

Isolation and identification of hyaluronan-degrading bacteria from Japanese fecal microbiota

PONE-D-23-02543R1

Dear Dr. Itsuko Fukuda,

We’re pleased to inform you that your manuscript has been judged scientifically suitable for publication and will be formally accepted for publication once it meets all outstanding technical requirements.

Kind regards,

Awatif Abid Al-Judaibi, PhD

Academic Editor

PLOS ONE

---

## [Editor Report · Acceptance letter]

9 May 2023

PONE-D-23-02543R1 

Isolation and identification of hyaluronan-degrading bacteria from Japanese fecal microbiota 

Dear Dr. Fukuda:

I'm pleased to inform you that your manuscript has been deemed suitable for publication in PLOS ONE. Congratulations! Your manuscript is now with our production department. 

Kind regards, 

on behalf of

Professor Awatif Abid Al-Judaibi 

Academic Editor

PLOS ONE